# Dietary Plant Sterols and Phytosterol-Enriched Margarines and Their Relationship with Cardiovascular Disease among Polish Men and Women: The WOBASZ II Cross-Sectional Study

**DOI:** 10.3390/nu14132665

**Published:** 2022-06-27

**Authors:** Anna Maria Witkowska, Anna Waśkiewicz, Małgorzata Elżbieta Zujko, Alicja Cicha-Mikołajczyk, Iwona Mirończuk-Chodakowska, Wojciech Drygas

**Affiliations:** 1Department of Food Biotechnology, Faculty of Health Sciences, Medical University of Bialystok, Szpitalna 37, 15-295 Bialystok, Poland; malgorzata.zujko@umb.edu.pl (M.E.Z.); iwona.mironczuk-chodakowska@umb.edu.pl (I.M.-C.); 2Department of Epidemiology, Cardiovascular Disease Prevention and Health Promotion, National Institute of Cardiology, Alpejska 42, 04-628 Warsaw, Poland; awaskiewicz@ikard.pl (A.W.); acicha@ikard.pl (A.C.-M.); wdrygas@ikard.pl (W.D.); 3Department of Social and Preventive Medicine, Faculty of Health Sciences, Medical University of Lodz, Hallera 1, 90-001 Lodz, Poland

**Keywords:** cardiovascular diseases, diabetes mellitus, humans, adult, phytosterols, diet, margarine

## Abstract

Dietary cholesterol has been suggested to increase the risk of cardiovascular disease (CVD). Phytosterols, present in food or phytosterol-enriched products, can reduce cholesterol available for absorption. The present study aimed to investigate the association between habitual intake of total and individual plant sterols (β-sitosterol, campesterol, and stigmasterol) or a diet combined with phytosterol-enriched products and CVD in a cross-section of Polish adults, participants of the Multicenter National Health Survey II (WOBASZ II). Among men (*n* = 2554), median intakes of plant sterols in terciles ranged between 183–456 mg/d and among women (*n* = 3136), 146–350 mg/d in terciles. The intake of phytosterols, when consumed with food containing phytosterols, including margarine, ranged between 184–459 mg/d for men and 147–352 mg/d for women. Among both men and women, beta-sitosterol intake predominated. Plant sterol intake was lower among both men and women with CVD (*p* = 0.016) compared to those without CVD. Diet quality, as measured by the Healthy Diet Index (HDI), was significantly higher in the third tercile of plant sterol intake for both men and women and the entire study group (*p* < 0.0001). This study suggests that habitual dietary intake of plant sterols may be associated with a lower chance of developing CVD, particularly in men.

## 1. Introduction

Cardiovascular disease (CVD) is a global health problem and a leading cause of death [1]. CVD risk factors are associated with poor lifestyle, including smoking, physical inactivity, obesity, unhealthy diet, and excessive alcohol consumption, leading to hypertension, hyperglycemia, and high LDL cholesterol [2,3]. Studies indicate a link between CVD and diabetes [3,4].

Type 2 diabetes mellitus (T2DM) predisposes patients to cardiovascular disease and cardiovascular mortality [5]. The development and progression of T2DM are strongly influenced by diet, physical inactivity, and increased body weight; therefore, intensive lifestyle modification is recommended for T2DM [6]. In patients with diabetes, the addition of soluble dietary fiber and phytosterols is recommended as a primary measure to prevent CVD before considering non-statin therapy [7].

Phytosterols (plant sterols and plant stanols) are natural bioactive plant substances with a structure similar to cholesterol. In the intestine, phytosterols and cholesterol compete for the same absorption mechanisms [8]. As a result, phytosterols can affect blood cholesterol concentrations by reducing the amount of cholesterol available for absorption. Studies have shown that consumption of 0.6–3.3 g of plant sterols per day reduces serum LDL-C concentrations by approximately 6–12%, and this effect was dose-dependent [9].

The diet typically provides 150–400 mg of plant sterols [10,11,12,13,14,15]. The phytosterols found in the highest amounts in plant-based foods, and, thus, in the human diet, are β-sitosterol, campesterol, and stigmasterol [16]. Food sources with the highest plant sterol content are vegetable oils, mainly corn oil, and sesame seeds [17]. Phytosterols isolated mainly from vegetable oils and their commercially produced esters can be ingredients of fortified foods and supplements as a non-pharmacological therapy of hypercholesterolemia. In European Union countries, products enriched in plant sterols are mainly milk and yogurt, margarine, and spreadable fats [18]. Plant sterol-enriched foods that provide 2 mg of phytosterols daily, combined with a healthy lifestyle, in patients with mild to moderate hypercholesterolemia have been found to reduce LDL-C levels by 10% [19,20]. However, the effect of long-term use of phytosterol-enriched foods on cardiovascular risk factors is unknown [21].

A few population-based studies, but not in the Polish population, have analyzed the effects of dietary phytosterol intake on CVD [10,11,14], but none included phytosterol-enriched products. Therefore, the present study aimed to investigate whether there is an association between habitual intake of total phytosterols and individual phytosterols (β-sitosterol, campesterol, and stigmasterol), or a diet combined with phytosterol-enriched products, and CVD in a cross-section of Polish adults.

## 2. Materials and Methods

### 2.1. Study Group

The study group consisted of 2554 men and 3136 women, of the National Multicenter Health Survey II (in Polish—WOBASZ II). WOBASZ II is a cross-sectional study representative of the Polish population of adults aged 20 years and older, which was conducted by the Institute of Cardiology (at present National Institute of Cardiology), Warsaw, Poland, in 2013–2014, in collaboration with five national medical universities. The design and methods of the WOBASZ II study have been described in detail elsewhere [22]. Approval for the WOBASZ II study was obtained from the Bioethics Committee at the National Institute of Cardiology (No. 1344), and was approved for the current study (No. 1837). Written informed consent was obtained from all participants.

Data on participants’ demographics, diseases, leisure-time physical activities, tobacco use, and alcohol intake, were collected using a standardized questionnaire developed for the WOBASZ II study. The classification of cardiovascular disease (CVD) was adopted according to World Health Organization guidelines [23]. Respondents were defined as having CVD if they had a reported history of any of the following: coronary heart disease, myocardial infarction, stroke, atrial fibrillation and/or other cardiac arrhythmias, peripheral vascular disease of the lower limbs, heart failure, coronary angioplasty or coronary artery bypass grafting, and implanted pacemaker or cardioverter-defibrillator. The criterion for diabetes, according to the American Diabetes Association [24], was a glucose level ≥ 7.0 mmol/L and/or use of glucose-lowering medication. Blood pressure (BP) was measured three times on the right arm after 5 min of rest in a sitting position at 1 min intervals, and the final BP was reported as the mean of the second and third measurements. Hypertension was diagnosed when systolic blood pressure was ≥140 mmHg and/or diastolic blood pressure ≥ 90 mmHg and/or when antihypertensive drugs were used. Height and weight measurements were taken by personnel trained in standard procedures. Body mass index (BMI) was calculated from body weight in kilograms divided by the square of height in meters. Biochemical analyses were performed at the Central Laboratory “Diagnostyka” at the National Institute of Cardiology in Warsaw.

### 2.2. Food Intake and Nutritional Assessment

Data on daily food intake were collected by trained interviewers using the single 24-h dietary recall method. To reduce the possibility of bias, individuals who described their diet as atypical were excluded. Based on the different types of food consumed, energy and dietary fiber of each patient’s diet were calculated using Polish food composition tables [25]. Polyphenols and antioxidants were calculated using previous studies [26,27,28,29].

### 2.3. Assessment of Healthy Diet Index (HDI) Score

Diet quality was determined by scoring the Healthy Diet Indicator (HDI), which was in accordance with the World Health Organization (WHO) dietary guidelines [18] and described in Fransen et al. [30]. HDI is based on six components—intake of saturated fatty acids (% total energy, %TE), intake of polyunsaturated fatty acids (%TE), dietary cholesterol (mg/d), dietary protein (%TE), fiber (g/d), and free sugars (%TE)—and fruits and vegetables (g/d), within the recommended range [31]. The final HDI score was the sum of all components, ranging from zero (minimal compliance with recommendations) to seven (maximum compliance with recommendations).

### 2.4. Assessment of Dietary Phytosterol Intake and the Intake of Plant Sterols from Enriched Margarine

Phytosterol intake was calculated as previously described using a developed database [12]. Total and individual phytosterol intakes were determined by multiplying the daily intake of each food by the total and individual phytosterol content of that food, respectively. Dietary recalls were reviewed by checking for consumption of phytosterol-enriched products. Based on the dietary history it was found that among the products enriched with phytosterols, only phytosterol-enriched margarine was consumed by 1.96% of men and 1.85% of women [12]. Manufacturers were identified and asked to report the plant sterol content of their products.

### 2.5. Statistical Analysis

The study population was divided into three groups according to the tercile distributions of plant sterol intakes (separately for total and individual phytosterols). All analyses were performed according to gender and overall. Quantitative variables were presented as mean (standard deviation) and/or median (interquartile range), while qualitative variables were presented as percentages. Mean values of plant sterol intake with a 95% confidence interval (95% CI), adjusted for age, were calculated using the general linear model and the Tukey-Kramer test was chosen for multiple comparisons, if appropriate. The odds ratios (ORs) with 95% CI for CVD were evaluated using logistic regression analysis in relation to total and particular phytosterol intake. Two models were applied: model 1, unadjusted in men and women but adjusted for sex, and combined, and model 2, adjusted for age, consumption of lipid-lowering drugs, HDI, BMI, alcohol intake, and, additionally, for sex, for the entire population. The first tercile (T1) in each model was adopted as a reference. Statistical analyses were carried out using SAS software version 9.4 (SAS Institute Inc., Cary, NC, USA). A *p*-value less than 0.05 was considered statistically significant.

## 3. Results

The general characteristics of the study participants are shown in Table 1. The mean age of the entire study group was 49.58 years. The highest percentage of the study participants had hypercholesterolemia 67.3% and hypertension 45.22%. CVD was diagnosed in 20% of the studied population, while diabetes in 10.82%.

Table 2 shows the phytosterol intake according to age, presence of diabetes, and CVD. The results are presented for men, women, and the entire study group. Dietary phytosterol content was found to be age-dependent and generally highest among the youngest age group and lowest among those aged 65 years and older. Among men and in the entire study group, sterol intake was significantly lower among people with diabetes (results were adjusted for age). No significant differences were found for women. With respect to CVD, plant sterol intake was lower among both men and women with CVD (*p* = 0.0016) and for both genders (*p* < 0.0001). With regard to diabetes, such a relationship was observed for men and the entire group, but not for women. With respect to individual plant sterols, we found that dietary intake of phytosterols was lower among both men with CVD and women and among men with diabetes (except campesterol in men with diabetes). No differences were found between women with diabetes and healthy women. For the whole group, only campesterol was not statistically significant. The intake of individual plant sterols with the fortified margarine was not considered, because manufacturers only reported the total phytosterol content. Thus, it was not possible to determine what the individual plant sterol content of the margarine was.

Table 3 shows terciles of plant sterol intake with food, and with food including phytosterol-enriched margarine. Terciles of individual plant sterol intake for the entire study group and by gender were used as means (crude, adjusted), medians, and ranges for particular phytosterols intake. Among men, the median plant sterol intake in the first tercile was 183, in the second tercile 292, and in the third tercile 456 mg/d. For food intake, including margarine with phytosterols, the values were 184; 294, and 459 mg/d, respectively. Among women, the median intakes of plant sterols with diet were: 146 in the first tercile, 231 in the second tercile, and 350 mg/d in the third tercile. For food intake, including margarine with phytosterols, these values for women were, respectively: 147; 232, and 352 mg/d. For individual plant sterols, they are ranked in Table 3 by the volume of intake. Among both men and women, beta-sitosterol intake predominated, with a median range of 112–280 mg/d per tercile among men and 91–222 mg/d among women. For campesterol, the median range was 31–107 mg/d among men and 24–78 mg/d among women, and for stigmasterol, 12–39 mg/d among men and 12–34 mg/d for women.

The odds ratio of developing CVD was related to phytosterol intake with diet (Figure 1). In the crude model, it was found that in both men and women, and in the entire study group (adjusted for gender), OR of CVD were significantly lower in the second and third terciles compared to the first terciles, with the lowest incidence of CVD in the third tercile. After adjusting for confounding factors, among men statistical significance was maintained, except for the second tercile of beta-sitosterol intake. Among women, only the intake of total plant sterols from the diet and their total intake together with margarine in the third tercile, and the intake of beta-sitosterol in the third tercile remained statistically significant. In the entire study group, significant values were observed in the third tercile of total plant sterol intake (without and with phytosterol-enriched products), and for all individual plant sterols.

We also investigated whether the results obtained could be biased by diet (Table 4). For this purpose, data from the extreme terciles of total and single plant sterol intake before and after energy adjustment were presented by sex and the entire study group. It was found that both before and after adjustment the results were significant for total and single dietary plant sterol. For intakes of phytosterol-enriched margarine, a significant difference was found between the first and third terciles before energy adjustment, which did not occur after adjustment. Intakes of polyphenols, antioxidants, dietary fiber, and HDI were also divided according to the tercile of total and individual plant sterol intake, with polyphenols, antioxidants, and dietary fiber adjusted for energy value. It was found that before adjustment, dietary polyphenol, antioxidant, and fiber contents were higher in the third tercile among both men and women and in the group as a whole (*p* < 0.0001). After adjustment for energy, differences were not observed. Diet quality, as measured by HDI, was significantly higher in the third tercile of plant sterol intake for both men and women and for the entire study group (*p* < 0.0001).

Intakes of atherogenic and antiatherogenic products were also examined in the first and third terciles of total and individual phytosterol intake (Table 5). For atherogenic products, butter and animal fat consumption was found to be higher in the third tercile of plant sterol intake, but after adjustment for energy there was an inverse difference, i.e., with higher plant sterol intake, animal fat and butter consumption was lower. For red meat and meat products before and after adjustment for energy, consumption was higher in the third tercile. All the above observations were true for both men and women and for the entire study group.

In the case of intake of antiatherogenic products, it was found that in both sexes and in the entire study group, both before and after adjustment for energy, the intake of vegetable oils, vegetable fats, fish, fruits, legumes, nuts, and seeds was higher in the third tercile of plant sterol intake. For soft margarine and vegetables, there were similar differences among men and the overall study group, but not among women. In women, after adjustment, differences were not observed. For whole grain bread, higher consumption by both sexes and in the entire study group was observed in the third tercile, but after adjusting for energy, differences were not significant.

## 4. Discussion

The prevalence of CVD and its risk factors among Poles is high [32]. CVD in this present population-based cross-sectional study was found in one fifth of the participants, which is concordant with the literature. This population requires interventions to reduce the incidence of CVD. One of the non-pharmacological treatment measures is a dietary modification to improve the quality of nutrition. Phytosterols, present in food and phytosterol-enriched food products, depending on the dose, can be effective in reducing LDL cholesterol, which is one of the risk factors for CVD [9].

Scientific evidence based on supplementation studies shows that the intake of 2 g of phytosterols is effective in lowering LDL cholesterol [20]. The relationship between dietary phytosterols and CVD is, however, controversial, as foods provide phytosterols in lower doses than dietary supplements do. The usual intake of phytosterols is generally less than 400 mg/day [10,11,12,13,14,15], and higher levels have been found only in vegans [33]. In this study, intakes higher than 400 mg/day were observed only in the highest tercile of phytosterol consumption, both in men and women. Previous evidence indicates, however, that phytosterols from natural foods may have an LDL cholesterol lowering effect [9]. In this study, both men and women with CVD were found to have lower intakes of total and individual plant sterols from diet and from diet and phytosterol-enriched margarine, than their healthy counterparts.

Diabetes predisposes one to CVD and people identified with diabetes are at a greater risk of developing cardiovascular diseases [5]. Scientific evidence shows that plant sterols can have beneficial effects on diabetes by reducing insulin resistance [34]. In this study, men with diabetes had significantly lower intakes of total and individual plant sterols, but no significant difference was observed in women.

Recent studies conducted in Poland support the belief that it is men who require special preventive measures to reduce cardiovascular risk factors, especially hypertension, dyslipidemia, diabetes, excessive body weight, and smoking [32,35]. Our cross-sectional study suggests that it is men who may benefit from habitual plant sterol intake. This is particularly evident after adjusting plant sterol intake for confounding variables, which were age, lipid-lowering medication, HDI, BMI, and alcohol. Among women, the findings are ambiguous because, after adjusting for confounders, most of the previously significant differences were not further observed for the second tercile of total and individual plant sterol intake. This might be due to the generally lower intake of plant sterols among women relative to that observed among men, and in the second quartile, it is low enough to observe beneficial effects. It is only in the third tercile of total and individual plant sterol intake that a lower incidence of CVD is observed among women.

The results of our study are in line with those of a Swedish study, which found that consumption of naturally occurring plant sterols was associated with a lower risk of a first heart attack in men, but not among women [10]. It is possible that women may benefit not from a single dietary component, but from a combination of foods and nutrients, which, for example, can be found in the Mediterranean diet or Dietary Portfolio [36]. Dietary recommendations to date regarding the consumption of a varied diet, and particularly emphasizing the consumption of plant-derived products, are reasonable in terms of providing various compounds of importance in the prevention of noncommunicable diseases. The contribution of phytosterols to the diet is highlighted by the Dietary Portfolio, which uses a combination of established nutritional approaches to lowering cholesterol, such as consumption of plant protein, nuts, soluble fiber, and monounsaturated fats and phytosterols [37,38]. It has been shown to improve LDL cholesterol fraction and other CVD risk factors [37,38,39]. In several other studies, lower levels of total cholesterol and LDL cholesterol were observed in relation to dietary phytosterols [40,41,42]. A recent study found that closer adherence to a plant-based diet was significantly associated with a lower risk of total CVD, coronary heart disease, and heart failure in postmenopausal women [36]. In contradiction to the Swedish study is the Danish study, which found no reduced CVD risk despite lower LDL-C concentrations in men [14]. However, the authors concluded that the study population had a narrow range of phytosterol intake.

Our study suggests that, in terms of intake of substances with beneficial effects on CVD, such as polyphenols, antioxidants, and dietary fiber, individuals with low and high intakes of plant sterols do not differ. However, they do differ in their intake of foods considered pro- and anti-atherogenic. It was found that study participants who had higher plant sterol intakes consumed more anti-atherogenic foods and fewer animal fats.

Phytosterol-enriched foods are recommended for people with hypercholesterolemia for the prevention of CVD [19]. However, in the WOBASZ II study, consumption of phytosterol-enriched foods was observed in a small proportion of the study group (less than 2% of participants). This translated into similar intakes of plant sterols and plant sterols along with phytosterols from fortified products.

### Limitations

The main limitation of the study was its cross-sectional nature, as a result of which causal inferences cannot be drawn. Another limitation was the use of a single 24-h recall method, which may not reflect the usual pattern of food consumption. To reduce the possibility of bias, subjects who described their diet as atypical were excluded.

The strengths of this study are its representativeness to the Polish population and the assessment of diet quality, which may act synergistically with plant sterols, which has not been studied before. To minimize the synergistic effect of plant sterols on the association with CVD, the results were adjusted for the Healthy Diet Index (HDI) score. A strength of the study was its consideration of phytosterol-enriched foods.

## 5. Conclusions

This study suggests that habitual dietary intake of plant sterols may be associated with a lower chance of developing CVD, particularly in men. However, this finding should be treated with caution because of the difficulty in separating the effects of plant sterols from the effects of other dietary components that may have synergistic effects.

## Figures and Tables

**Figure 1 nutrients-14-02665-f001:**
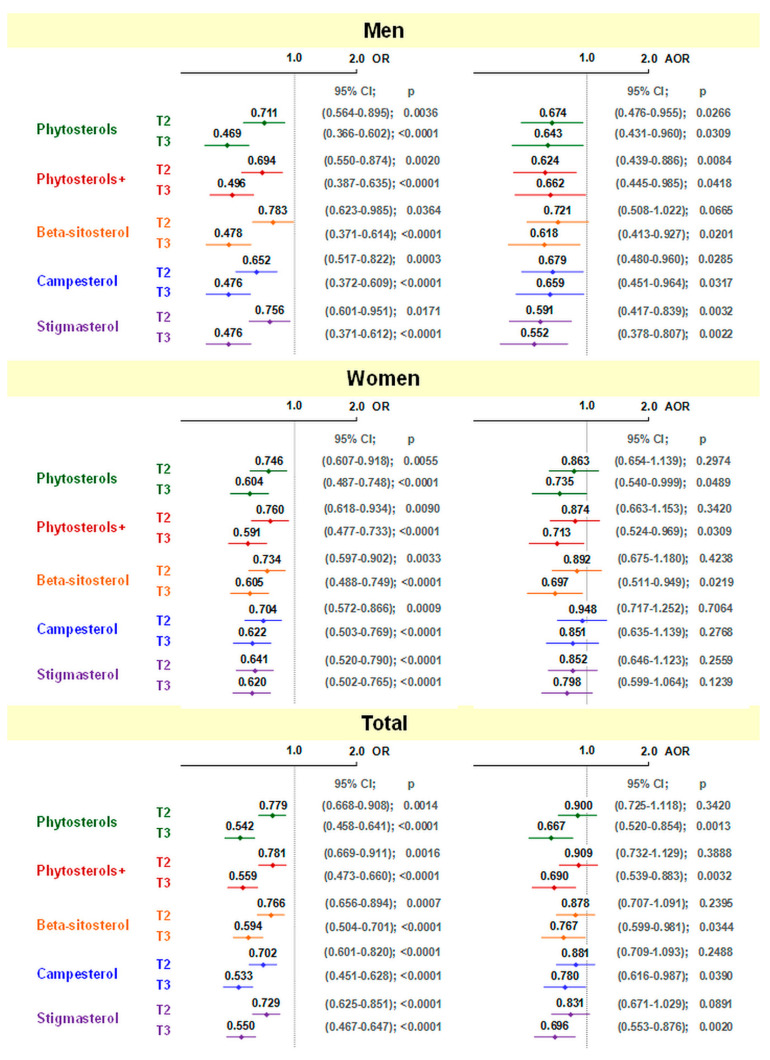
Odds ratio (95% confidence interval) for CVD in relation to total and individual phytosterol intake (relative to 1st tercile). T2—2nd tercile; T3—3rd tercile; OR—odds ratio; AOR—adjusted odds ratio; ORs were unadjusted in men and women but adjusted for sex combined; AORs—adjusted for age, lipid-lowering medication, HDI, BMI, alcohol, and additionally for sex in total.

**Table 1 nutrients-14-02665-t001:** General characteristics of the studied population.

Trait	Men *n* = 2554	Women *n* = 3136	Total *n* = 5690
Age (year), mean ± SD	48.79 ± 16.27	50.23 ± 16.54	49.58 ± 16.43
BMI (kg/m^2^), mean ± SD	27.42 ± 4.55	26.96 ± 5.65	27.17 ± 5.19
CVD, (%)	19.34	20.54	20.00
Hypertension, (%) ^1^	49.56	41.69	45.22
Hypercholesterolemia, (%) ^2^	68.86	66.03	67.30
Diabetes, (%) ^3^	11.86	9.96	10.82

^1^ Hypertension: systolic blood pressure SBP ≥ 140 mmHg or diastolic blood pressure DBP ≥ 90 mmHg, or use of antihypertensive drugs. ^2^ Hypercholesterolemia: total cholesterol ≥ 5 mmol/L or LDL cholesterol ≥ 3 mmol/L or the participant was taking lipid-lowering medication. ^3^ Diabetes: blood glucose level was ≥7.0 mmol/L or diabetes was declared in an interview.

**Table 2 nutrients-14-02665-t002:** Phytosterol intakes depending on age, diabetes, and CVD.

	Men		Women		Total	
	*n* = 2554		*n* = 3136		*n* = 5690	
	Mean (95% CI)	*p*	Mean (95% CI)	*p*	Mean (95% CI)	*p*
Total plant sterol intake from food (mg/d)						
Age (years)						
20–44	344.3 (333.0–351.9)	<0.0001	258.5 (251.7–265.2)	<0.0001	300.0 (294.3–305.6)	<0.0001
45–64	323.0 (313.1–332.8)		263.1 (256.5–269.7)		293.5 (287.8–299.2)	
65+	264.1 (249.7–278.4)		217.8 (208.5–227.1)		242.5 (234.3–250.7)	
Diabetes (mg/d)						
no	323.3 (316.6–330.1)	0.0379	253.6 (249.0–258.2)	0.1289	288.4 (284.4–292.4)	0.0077
yes	301.7 (282.6–320.8)		241.9 (227.5–256.2)		271.3 (259.6–283.1)	
CVD (mg/d)						
no	326.1 (319.1–333.1)	0.0016	255.9 (251.1–260.7)	0.0016	290.9 (286.8–295.0)	<0.0001
yes	298.6 (283.5–313.6)		237.8 (227.9–247.7)		268.3 (259.7–277.0)	
Total phytosterols intake from food and enriched margarine (mg/d)						
Age (years)						
20–44	344.3 (334.7–353.9)	<0.0001	258.9 (252.1–265.7)	<0.0001	301.1 (295.4–306.9)	<0.0001
45–64	326.0 (316.0–336.0)		264.9 (258.3–271.6)		295.9 (290.1–301.7)	
65+	266.6 (252.0–281.2)		219.6 (210.2–229.0)		244.8 (236.4–253.1)	
Diabetes (mg/d)						
no	325.3 (318.4–332.2)	0.0789	255.1 (250.4–259.7)	0.0941	290.1 (286.1–294.2)	0.0121
yes	306.6 (287.2–326.1)		242.0 (227.5–256.4)		273.9 (261.9–285.8)	
CVD (mg/d)						
no	328.2 (321.1–335.3)	0.0034	257.2 (252.4–262.1)	0.0016	292.7 (288.5–296.9)	<0.0001
yes	302.3 (286.9–317.6)		238.9 (228.9–248.9)		270.6 (261.8–279.4)	
Beta-sitosterol (mg/d)						
Age (years)						
20–44	209.7 (204.0–215.5)	<0.0001	160.9 (156.8–165.0)	<0.0001	185.0 (181.5–188.5)	<0.0001
45–64	200.1 (194.1–206.2)		165.9 (161.9–170.0)		183.3 (179.8–186.8)	
65+	164.5 (155.7–173.3)		137.0 (131.3–142.6)		151.6 (146.6–156.6)	
Diabetes (mg/d)						
no	199.3 (195.2–203.4)	0.0468	159.1 (156.3–161.9)	0.0840	179.2 (176.7–181.6)	0.0065
yes	186.6 (175.0–198.3)		150.9 (142.2–159.7)		168.5 (161.3–175.7)	
CVD (mg/d)						
no	200.7 (196.4–205.0)	0.0062	160.4 (157.5–163.3)	0.0016	180.5 (178.0–183.1)	<0.0001
yes	186.1 (176.9–195.3)		149.4 (143.3–155.4)		167.7 (162.4–173.0)	
Campesterol						
Age (years)						
20–44	75.6 (72.8–78.4)	<0.0001	52.6 (50.7–54.5)	<0.0001	63.9 (62.3–65.6)	<0.0001
45–64	68.4 (65.5–71.4)		52.9 (51.0–54.7)		60.8 (59.1–62.5)	
65+	55.3 (51.0–59.6)		43.3 (40.7–45.9)		49.7 (47.4–52.1)	
Diabetes (mg/d)						
no	69.5 (67.5–71.5)	0.3479	50.9 (49.7–52.2)	0.6694	60.2 (59.1–61.4)	0.2715
yes	66.6 (60.9–72.3)		50.0 (46.1–54.0)		58.2 (54.8–61.6)	
CVD (mg/d)						
no	70.6 (68.5–72.6)	0.0044	51.9 (50.5–53.2)	0.0015	61.2 (60.0–62.4)	<0.0001
yes	63.2 (58.7–67.7)		46.9 (44.1–49.6)		55.1 (52.6–57.5)	
Stigmasterol						
Age (years)						
20–44	28.0 (27.0–28.9)	<0.0001	24.1 (23.4–24.8)	<0.0001	26.0 (25.4–26.6)	<0.0001
45–64	27.0 (26.0–28.0)		24.5 (23.8–25.2)		25.8 (25.2–26.3)	
65+	21.2 (19.8–22.7)		19.0 (18.0–20.0)		20.2 (19.4–21.0)	
Diabetes (mg/d)						
no	26.8 (26.2–27.5)	0.0007	23.4 (22.9–23.9)	0.1717	25.1 (24.7–25.5)	0.0004
yes	23.3 (21.4–25.2)		22.3 (20.8–23.8)		22.8 (21.6–24.0)	
CVD (mg/d)						
no	26.9 (26.2–27.6)	0.0017	23.6 (23.1–24.1)	0.0023	25.2 (24.8–25.7)	<0.0001
yes	24.2 (22.6–25.7)		21.7 (20.7–22.8)		23.0 (22.1–23.9)	

Results adjusted for age in men and women and additionally for sex in total; adjustment not applicable to the age groups.

**Table 3 nutrients-14-02665-t003:** Intake of phytosterols in terciles.

	Men			Women			Total		
	N = 2554			N = 3136			N = 5690		
	Tercile 1	Tercile 2	Tercile 3	Tercile 1	Tercile 2	Tercile 3	Tercile 1	Tercile 2	Tercile 3
	*n* = 851	*n* = 851	*n* = 852	*n* = 1045	*n* = 1045	*n* = 1046	*n*=1896	*n*=1897	*n*=1897
Total phytosterol intake from food (mg/d)									
Mean ± SD	173.8 ± 43.9	292.3 ± 32.8	496.0 ± 146.9	140.6 ± 33.6	233.1 ± 26.1	382.7 ± 111.4	152.8 ± 38.1	257.2 ± 29.9	438.8 ± 135.7
Me (IQR)	183.2 (142.0–210.0)	291.6 (265.1–319.6)	455.9 (398.7–547.1)	146.4 (119.2–167.7)	230.6 (210.0–254.4)	349.9 (308.1–418.5)	158.0 (127.0–185.0)	255.9 (229.8–283.4)	399.8 (348.0–487.9)
Range	0.23–234.7	234.7–353.6	354.0–1774.0	27.2–190.2	190.3–281.7	281.8–1632.7	0.23–207.8	207.8–310.6	310.7–1774.0
Adjusted mean (95% CI) *	174.8 (168.7–180.9)	292.5 (286.5–298.6)	494.8 (488.7–500.9)	140.9 (136.7–145.1)	233.1 (228.9–237.2)	382.5 (378.3–386.6)	156.6 (152.8–160.4)	258.4 (254.6–262.1)	436.4 (432.7–440.2)
Total phytosterol intake from food and enriched margarine (mg/d)									
Mean ± SD	174.2 ± 44.0	294.0 ± 33.3	501.1 ± 150.9	141.0 ± 33.9	234.2 ± 26.2	385.1 ± 112.1	153.2 ± 38.3	258.4 ± 30.1	442.6 ± 138.4
Me (IQR)	183.9 (142.5–210.4)	294.0 (266.7–321.0)	459.1 (403.8–550.6)	146.9 (119.2–168.4)	232.2 (211.3–255.5)	352.0 (309.6–422.5)	158.9 (127.1–185.4)	257.2 (230.9–284.6)	404.5 (350.3–491.1)
Range	0.23–235.2	235.3–356.2	356.4–1774.0	27.2–191.2	191.2–283.1	283.1–1632.7	0.23–208.6	208.8–312.1	312.1–1774.0
Adjusted mean (95% CI) *	175.2 (168.9–181.4)	294.3 (288.0–300.5)	500.0 (493.7–506.3)	141.2 (137.0–145.4)	234.2 (230.0–238.4)	384.9 (380.7–389.1)	157.0 (153.1–160.8)	259.6 (255.8–263.4)	440.2 (436.4–444.1)
B-sitosterol (mg/d)									
Mean ±SD	107.3 ± 27.9	181.1 ± 20.3	305.2 ± 88.1	87.6 ± 22.0	147.0 ± 16.3	239.7 ± 62.7	95.0 ± 24.6	160.9 ± 18.3	272.0 ± 79.3
Me (IQR)	112.4 (87.1–131.2)	180.8 (163.9–197.8)	279.6 (246.9–332.9)	90.5 (72.8–105.9)	146.2 (133.0–161.4)	222.0 (196.1–264.3)	98.35 (77.4–115.8)	160.8 (144.4–176.6)	248.2 (218.1–301.3)
Range	0.20–146.0	146.0–219.3	219.3- 1028.7	14.8–120.0	120.0–175.9	176.0–679.6	0.20–130.7	130.8–194.6	194.7–1028.7
Adjusted mean (95% CI) *	107.8 (104.1–111.4)	181.2 (177.5–184.8)	304.6 (300.9–308.3)	87.8 (85.4–90.2)	146.9 (144.5–149.3)	239.6 (237.2–242.0)	97.2 (95.0–99.4)	161.5 (159.3–163.7)	270.8 (268.6–273.0)
Campesterol (mg/d)									
Mean ±SD	30.2 ± 8.6	57.2 ± 8.5	120.0 ± 49.3	23.0 ± 6.0	41.7 ± 6.1	87.8 ± 32.3	25.5 ± 7.1	48.2 ± 7.8	103.4 ± 42.8
Me (IQR)	31.4 (23.9–37.2)	56.7 (49.7–64.1)	106.6 (86.4–137.8)	23.7 (18.7–27.8)	40.9 (36.4–46.4)	78.1 (64.9–100.8)	26.1 (20.6–31.5)	47.4 (41.4–54.4)	90.7 (74.4–118.5)
Range	0.02–43.5	43.5–73.8	73.9–585.8	3.2- 32.2	32.2–54.2	54.3–342.3	0.02–36.3	36.3–63.5	63.5–585.8
Adjusted mean (95% CI) *	30.6 (28.6–32.6)	57.2 (55.2–59.1)	119.7 (117.7–121.6)	23.0 (21.8–24.2)	41.7 (40.6–42.9)	87.7 (86.6–88.9)	26.7 (25.5–27.9)	48.5 (47.3–49.6)	102.9 (101.7–104.0)
Stigmasterol (mg/d)									
Mean ±SD	11.9 ± 3.9	23.6 ± 3.2	43.6 ± 15.3	11.2 ± 3.5	21.2 ± 2.7	37.3 ± 11.6	11.5 ± 3.7	22.2 ± 2.9	40.2 ± 13.6
Me (IQR)	12.2 (9.1–15.1)	23.5 (20.7–26.4)	39.1 (32.9–48.1)	11.7 (8.7–14.2)	21.1 (18.8–23.6)	33.9 (29.2–41.8)	11.8 (8.9–14.5)	22.1 (19.6–24.7)	36.1 (30.9–44.9)
Range	0.003–18.3	18.3–29.2	29.3–131.7	0.9–16.4	16.4–26.0	26.1–132.3	0.003–17.2	17.2–27.3	27.4–132.3
Adjusted mean (95% CI) *	12.0 (11.4–12.6)	23.6 (23.0–24.2)	43.5 (42.8–44.1)	11.3 (10.8–11.7)	21.1 (20.7–21.6)	37.3 (36.8–37.7)	11.6 (11.3–12.0)	22.3 (21.9–22.6)	40.1 (39.8–40.5)

* Results were adjusted for age in men and women and additionally for sex in total.

**Table 4 nutrients-14-02665-t004:** Dietary quality in relation to tercile of dietary total and individual phytosterol intake (1st tercile vs. 3rd tercile).

	Men			Women			Total		
	N = 2554			N = 3136			N = 5960		
	1st Tercile	3rd Tercile	*p*	1st Tercile	3rd Tercile	*p*	1st Tercile	3rd Tercile	*p*
	*n* = 851	*n* = 852		*n* = 1045	*n* = 1046		*n* = 1896	*n* = 1897	
	Mean (95% CI)	Mean (95% CI)		Mean (95% CI)	Mean (95% CI)		Mean (95% CI)	Mean (95% CI)	
Phytosterols									
Total plant sterols (mg/d)	174.8 (168.7–180.9)	494.8 (488.7–500.9)	<0.0001	140.9 (136.7–145.1)	382.5 (378.3–386.6)	<0.0001	156.6 (152.8–160.4)	436.4 (432.7–440.2)	<0.0001
Total plant sterols/1000 kcal (mg/d)	111.6 (108.8–114.4)	171.8 (169.1–174.6)	<0.0001	124.1 (121.2–127.1)	189.7 (186.7–192.6)	<0.0001	117.9 (115.9–120.0)	180.9 (178.8–183.0)	<0.0001
B-sitosterol (mg/d)	108.4 (104.6–112.1)	303.9 (300.1–307.7)	<0.0001	88.6 (86.1–91.1)	238.5 (236.1–241.0)	<0.0001	97.6 (95.3–99.9)	269.7 (267.4–271.9)	<0.0001
B-sitosterol/1000 kcal (mg/d)	69.2 (67.5–71.0)	105.7 (103.9–107.4)	<0.0001	78.1 (76.2–79.9)	118.2 (116.4–120.1)	<0.0001	72.8 (71.4–74.1)	111.8 (110.5–113.1)	<0.0001
Campesterol (mg/d)	32.6 (30.4–34.8)	115.5 (113.3–117.7)	<0.0001	24.5 (23.1–25.8)	84.2 (82.8–85.5)	<0.0001	28.2 (26.9–29.5)	98.9 (97.6–100.1)	<0.0001
Campesterol/1000 kcal (mg/d)	20.7 (19.8–21.6)	40.1 (39.2–41.0)	<0.0001	21.4 (20.6–22.3)	42.1 (41.2–42.9)	<0.0001	20.5 (19.9–21.1)	41.0 (40.3–41.6)	<0.0001
Stigmasterol (mg/d)	16.2 (15.3–17.1)	37.6 (36.7–38.5)	<0.0001	14.9 (14.2–15.5)	31.4 (30.8–32.1)	<0.0001	15.6 (15.1–16.2)	34.4 (33.9–35.0)	<0.0001
Stigmasterol/1000 kcal (mg/d)	10.7 (10.3–11.2)	13.2 (12.7–13.7)	<0.0001	13.5 (13.0–14.1)	15.6 (15.1–16.1)	<0.0001	12.2 (11.9–12.6)	14.4 (14.0–14.7)	<0.0001
Phytosterols from enriched margarine (mg/d)	1.03 (0.00–2.52)	3.81 (2.32–5.30)	0.0278	0.66 (0.02–1.31)	2.05 (1.41–2.70)	0.0083	0.80 (0.03–1.57)	2.93 (2.17–3.69)	0.0004
Phytosterols from enriched margarine/1000 kcal (mg/d)	0.69 (0.03–1.35)	1.42 (0.76–2.09)	0.2832	0.59 (0.20–0.99)	1.09 (0.70–1.49)	0.1861	0.57 (0.19–0.95)	1.25 (0.88–1.62)	0.0330
Polyphenols and antioxidants									
Polyphenols (mg/d)	1464 (1410–1518)	2705 (2651–2759)	<0.0001	1427 (1383–1472)	2494 (2450–2538)	<0.0001	1443 (1408–1478)	2594 (2559–2629)	<0.0001
Polyphenols/1000 kcal (mg/d)	945 (916–973)	945 (916–973)	0.9999	1260 (1224–1296)	1227 (1191–1263)	0.4125	1112 (1088–1136)	1086 (1062–1110)	0.2963
Antioxidants (mmol/d)	8.71 (8.25–9.18)	16.20 (15.74–16.67)	<0.0001	8.84 (8.43–9.25)	15.36 (14.95–15.77)	<0.0001	8.79 (8.48–9.11)	15.78 (15.47–16.09)	<0.0001
Antioxidants/1000 kcal (mmol/d)	5.70 (5.47–5.93)	5.71 (5.48–5.94)	0.9979	7.87 (7.57–8.17)	7.60 (7.29–7.90)	0.4324	6.86 (6.66–7.07)	6.66 (6.46–6.86)	0.3449
Dietary fiber									
Dietary fiber (g/d)	14.9 (14.4–15.4)	27.5 (27.0–28.0)	<0.0001	12.4 (12.0–12.8)	22.5 (22.1–22.9)	<0.0001	13.5 (13.2–13.9)	24.9 (24.6–25.2)	<0.0001
Dietary fiber/1000 kcal (g/d)	9.4 (9.1–9.6)	9.6 (9.4–9.8)	0.3899	10.7 (10.4–11.0)	11.1 (10.8–11.4)	0.1432	10.0 (9.9–10.2)	10.3 (10.1–10.5)	0.1132
Diet quality									
HDI score (points)	2.67 (2.59–2.75)	3.83 (3.75–3.92)	<0.0001	2.76 (2.69–2.83)	3.85 (3.78–3.91)	<0.0001	2.70 (2.65–2.76)	3.83 (3.77–3.88)	<0.0001

Results were adjusted for age in men and women and additionally for gender in men and women overall.

**Table 5 nutrients-14-02665-t005:** Consumption of selected products by tercile of phytosterol intake (1st tercile vs. 3rd tercile).

	Men			Women			Total		
	*n* = 2554			*n* = 3136			*n* = 5960		
	1st Tercile	3rd Tercile	*p*	1st Tercile	3rd Tercile	*p*	1st Tercile	3rd Tercile	*p*
	*n* = 851	*n* = 852		*n* = 1045	*n* = 1046		*n* = 1896	*n* = 1897	
	Mean (95% CI)	Mean (95% CI)		Mean (95% CI)	Mean (95% CI)		Mean (95% CI)	Mean (95% CI)	
Atherogenic food									
Butter (g/d)	13.1 (11.7–14.5)	16.9 (15.5–18.3)	0.0007	10.3 (9.4–11.2)	12.7 (11.8–13.6)	0.0005	11.8 (11.0–12.6)	14.9 (14.1–15.7)	<0.0001
Butter/1000 kcal (g/d)	7.8 (7.3–8.4)	5.3 (4.8–5.8)	<0.0001	8.2 (7.7–8.7)	5.8 (5.3–6.4)	<0.0001	8.1 (7.7–8.5)	5.6 (5.2–6.0)	<0.0001
Red meat and cold cuts (g/d)	108.3 (97.8–118.9)	214.2 (203.7–224.8)	<0.0001	53.6 (48.2–58.9)	101.9 (96.5–107.3)	<0.0001	84.1 (78.3–89.8)	156.4 (150.7–162.0)	<0.0001
Red meat and cold cuts/1000 kcal (g/d)	61.4(57.4–65.3)	69.7(65.8–73.6)	0.0102	41.0 (38.0–44.1)	48.1 (45.1–51.2)	0.0034	51.7 (49.2–54.2)	58.7 (56.3–61.2)	0.0003
Animal fats (g/d)	23.2 (21.2–25.3)	28.2 (26.1–30.2)	0.0024	18.6 (17.3–19.8)	20.4 (19.1–21.6)	0.1120	21.1 (19.9–22.2)	24.5 (23.4–25.7)	0.0001
Animal fats/1000 kcal (g/d)	13.4 (12.7–14.1)	8.7 (8.0–9.4)	<0.0001	14.4 (13.8–15.1)	9.1 (8.5–9.8)	<0.0001	14.0 (13.6–14.5)	9.0 (8.5–9.5)	<0.0001
Antiatherogenic food									
Oils (g/d)	2.8 (1.8–3.9)	25.3 (24.3–26.4)	<0.0001	1.9 (1.2–2.6)	19.4 (18.8–20.1)	<0.0001	2.2 (1.5–2.8)	22.0 (21.4–22.6)	<0.0001
Oils/1000 kcal (g/d)	1.9 (1.5–2.4)	8.9 (8.5–9.3)	<0.0001	1.6 (1.3–2.0)	10.0 (9.6–10.4)	<0.0001	1.5 (1.2–1.8)	9.3 (9.0–9.6)	<0.0001
Soft margarine (g/d)	6.9 (5.8–8.1)	18.3 (17.1–19.4)	<0.0001	4.7 (4.1–5.4)	9.8 (9.1–10.4)	<0.0001	6.0 (5.4–6.7)	13.9 (13.3–14.6)	<0.0001
Soft margarine/1000 kcal (g/d)	4.5 (4.0–5.0)	6.4 (5.9–6.9)	<0.0001	4.4 (4.0–4.8)	4.8 (4.4–5.2)	0.3933	4.5 (4.2–4.8)	5.6 (5.3–5.9)	<0.0001
Vegetable fats (g/d)	9.7 (8.3–11.2)	43.7 (42.3–45.1)	<0.0001	6.6 (5.8–7.5)	29.3 (28.4–30.1)	<0.0001	8.2 (7.3–9.0)	36.0 (35.2–36.8)	<0.0001
Vegetable fats/1000 kcal (g/d)	6.4 (5.8–7.0)	15.3 (14.7–15.9)	<0.0001	6.0 (5.5–6.6)	14.8 (14.3–15.3)	<0.0001	6.0 (5.6–6.4)	15.0 (14.6–15.4)	<0.0001
Fish (g/d)	11.7(7.1–16.3)	37.3(32.7–41.9)	<0.0001	7.6 (4.4–10.7)	25.7 (22.5–28.9)	<0.0001	8.8 (6.0–11.6)	31.3 (28.6–34.0)	<0.0001
Fish/1000 kcal (g/d)	7.5 (5.5–9.5)	13.3 (11.3–15.4)	0.0002	6.5 (4.5–8.4)	13.2 (11.2–15.2)	<0.0001	6.4 (4.9–7.8)	13.4 (11.9–14.8)	<0.0001
Wholemeal bread (g/d)	22.0 (17.6–26.5)	34.0 (29.5–38.5)	0.0007	18.5 (15.5–21.6)	32.4 (29.4–35.5)	<0.0001	19.2 (16.5–21.8)	32.6 (29.9–35.2)	<0.0001
Wholemeal bread/1000 kcal (g/d)	14.3 (12.1–16.4)	13.1 (10.9–15.2)	0.7170	17.8 (15.8–19.9)	16.5 (14.5–18.6)	0.6651	15.7 (14.2–17.2)	14.6 (13.1–16.1)	0.5920
Vegetables (g/d)	195 (183–208)	316 (304–328)	<0.0001	168 (159–178)	287 (277–296)	<0.0001	181 (173–188)	301 (293–309)	<0.0001
Vegetables/1000 kcal (g/d)	131 (124–138)	113 (106–120)	0.0013	153 (145–160)	146 (139–154)	0.4727	142 (137–147)	129 (124–135)	0.0024
Fruit (g/d)	119 (103–135)	268 (253–284)	<0.0001	134 (120–147)	303 (290–317)	<0.0001	123 (112–133)	282 (272–292)	<0.0001
Fruit/1000 kcal (g/d)	77 (69–84)	95 (87–102)	0.0018	119 (111–128)	149 (141–158)	<0.0001	98 (92–104)	121 (115–126)	<0.0001
Legume (g/d)	0.73(0.00–1.88)	7.73 (6.58–8.88)	<0.0001	0.77 (0.00–1.58)	5.64 (4.83–6.46)	<0.0001	0.85 (0.16–1.55)	6.74 (6.05–7.43)	<0.0001
Legume/1000 kcal (g/d)	0.53 (0.00–1.07)	3.00 (2.46–3.54)	<0.0001	0.79 (0.23–1.35)	3.07 (2.51–3.64)	<0.0001	0.69 (0.28–1.09)	3.07 (2.68–3.47)	<0.0001
Nuts (g/d)	0.60 (0.00–1.34)	2.92 (2.18–3.67)	<0.0001	0.21 (0.00–0.92)	2.91 (2.20–3.62)	<0.0001	0.21 (0.00–0.74)	3.10 (2.58–3.62)	<0.0001
Nuts/1000 kcal (g/d)	0.35 (0.04–0.65)	1.05 (0.74–1.36)	0.0045	0.14 (0.00–0.49)	1.34 (0.99–1.69)	<0.0001	0.14 (0.00–0.38)	1.29 (1.06–1.53)	<0.0001
Seeds (g/d)	0.11 (0.00–0.63)	1.34 (0.82–1.86)	0.0033	0.12 (0.00–0.40)	1.13 (0.85–1.40)	<0.0001	0.11 (0.00–0.39)	1.20 (0.92–1.48)	<0.0001
Seeds/1000 kcal (g/d)	0.10 (0.00–0.32)	0.51 (0.29–0.73)	0.0285	0.11 (0.00–0.27)	0.57 (0.41–0.74)	0.0003	0.10 (0.00–0.24)	0.54 (0.40–0.67)	<0.0001

Results were adjusted for age in men and women and additionally for sex in total.

## Data Availability

All data in this study are made available upon request to the authors at the following e-mail address: anna.witkowska@umb.edu.pl or awaskiewicz@ikard.pl.

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
