# Peer review of "Dietary Plant Sterols and Phytosterol-Enriched Margarines and Their Relationship with Cardiovascular Disease among Polish Men and Women: The WOBASZ II Cross-Sectional Study"

_nutrients, 2022, doi:10.3390/nu14132665_

Round 1
Reviewer 1 Report
This is a clinical study, which aimed to investigate whether there was an association between habitual intake of total phytosterols and individual phytosterols (β-sitosterol, campesterol, and stigmasterol) or a diet combined with phytosterol-enriched products and CVD in a cross-section of Polish adults. In conclusion, the authors suggested that habitual dietary intake of plant sterols might be associated with a lower chance of developing CVD, particularly in men.
This is clinically important; however, this reviewer considers that this paper has a big limitation. This reviewer has some criticisms as described below.
Major comments:
1. As the authors indicated in the Limitation section, this is cross-sectional, which has a big limitation to examine the development of CVD. Through the whole manuscript, the authors examined the association between CVD development and dietary plant sterols. To examine this hypothesis, they have to perform the longitudinal study. In current situation, they are not able to reach their conclusion.
2. The authors included CVD patients, and they examined the food intake data after CVD development. Thus, they collected the data of the patients’ current situation of food intake. Diet therapy should be performed after CVD development. How do the authors exclude the food intake changes after CVD development?
3. In conclusion, the authors described that plant sterols lower chance of developing CVD. However, as this reviewer described above, the authors are not able to reach this conclusion, unless they enrolled patients before CVD development and perform the longitudinal study.
Author Response
On behalf of the authors, thank you for reviewing this manuscript and for sharing your doubts regarding methodology. The authors again consulted with statisticians and there is no doubt that the study design is methodologically sound. Below you will find our explanations.
We would like to clarify that the WOBASZ 2 study was a cross-sectional study of the representative population of the adult Polish population. To date, there have been no studies in the Polish population that evaluated diet. This is a unique study because the method of selecting participants is based on the structure of the Polish population. The authors are aware of the fact that cross-sectional studies have their limitations, and they know that such studies are useful for establishing preliminary evidence for a causal relationship. Cross-sectional studies are also useful for examining the association between exposure and disease onset for chronic diseases where researchers lack information on time of onset. Examples might include diet and arthritis, smoking and chronic bronchitis, and asthma and exposure to air pollution. Interpretation requires caution regarding potential association of duration of disease with exposure status. Cross-sectional designs also are commonly used in population-based surveys to obtain information about the prevalence of outcomes that will be used to plan cohort studies. Thus, our study sets a certain direction for further proceedings and is particularly valuable because no such studies have been conducted in Poland so far.
In cross-sectional studies, it is possible to assess relationships (not the influence, not the causal effect) between the assessed parameters, provided that appropriate, highly advanced statistical methods are used. For this reason the logistic regression analysis was used, which is an essential research tool for cross-sectional studies. Logistic regression in cross-sectional studies enables to examine associations between variables. Thus, the chance (odds ratio, OR) of CVD occurrence in terciles of phytosterol consumption was presented in the manuscript. In addition, the type of study was included in the title of the manuscript, and limitations were included in the discussion to avoid confusion in interpreting the results of this study.
We agree with the reviewer's suggestion that the conclusions should be confirmed by longitudinal studies. In any case, we have emphasized this necessity in the discussion section of the manuscript. Consequently, conclusions were formulated very cautiously because we could only examine relationships, not causal effects. We also want to let you know that a representative sample of WOBASZ 2 participants is under long-term follow-up. We are going to prepare a manuscript on this topic after the project is completed.
Reviewer 2 Report
Dear Editor and Authors,
Regarding the manuscript "Dietary Intake Of Plant Sterols And Phytosterol-Enriched Margarines And Their Relationship With Cardiovascular Disease Among Polish Men And Women: Results Of The WOBASZ II Cross-Sectional Study" submitted to Nutrients journal, the following issues should be mentioned:
1. the title of the manuscript contain too many characters
2. the Abstract could contain the essential information.
3. the keyword could be written according to MeSH on Demand https://meshb.nlm.nih.gov/MeSHonDemand?_gl=1*1g9ykwp*_ga*MTA2NjUzOTE0MC4xNjIwMzc5MjEw*_ga_P1FPTH9PL4*MTY1NTIxMDgwMC40LjAuMTY1NTIxMDgwMC4w
4. the aim of the study is mentioned differently in different sections: CVD risk factors and CVD (alone)
5. according to which guideline was diabetes diagnosed?
6. the Material and Methods chapter contains the description of the study participants, it should be moved in the Results chapter
7. the Discussion chapter could be enriches with more data from the literature since it contains less information compared to Results chapter
8. the Conclusion of this observational study overestimates the results; since it is an observational study, no causality relationship can be mentioned; the speculation of causality can be mentioned in the Discussion chapter, but not in the Conclusion chapter.
Author Response
On behalf of the authors, thank you for the insightful and valuable review that will undoubtedly raise the level of our work.
- the title of the manuscript contain too many characters
The title of the manuscript has been modified to limit the number of characters. Further abbreviation of the title may not convey relevant information. Redundant information has been removed without loss of meaning.
- the Abstract could contain the essential information
The abstract has been revised to be more concise and to include the essential information.
- the keyword could be written according to MeSH on Demand
Thank you for suggesting the use of MESH on Demand. We used this tool to improve the keywords.
- the aim of the study is mentioned differently in different sections: CVD risk factors and CVD (alone)
Actually, there is no difference between the purpose of the paper in the abstract and in the Introduction section. However, to make the content clearer, any wording that could create ambiguity has been removed.
- according to which guideline was diabetes diagnosed?
Criteria for diagnosing diabetes according to American Diabetes Association (2019) have been added.
- the Material and Methods chapter contains the description of the study participants, it should be moved in the Results chapter
The description of the study participants was moved to the Results section. Consequently, we added a description of Table 1.
- the Discussion chapter could be enriches with more data from the literature since it contains less information compared to Results chapter.
Thank you for this apt comment. We have expanded the discussion.
- the Conclusion of this observational study overestimates the results; since it is an observational study, no causality relationship can be mentioned; the speculation of causality can be mentioned in the Discussion chapter, but not in the Conclusion chapter.
Thank you for this comment. We have removed the speculative sentence from the Conclusions.
Round 2
Reviewer 1 Report
This reviewer has no further comment.
Reviewer 2 Report
Dear Editor and Authors,
I have no further comments.